# Clinical Application of Platelet Concentrates in Bovine Practice: A Systematic Review

**DOI:** 10.3390/vetsci10120686

**Published:** 2023-12-01

**Authors:** Chiara Caterino, Giovanni Della Valle, Federica Aragosa, Stefano Cavalli, Jacopo Guccione, Francesco Lamagna, Gerardo Fatone

**Affiliations:** Department of Veterinary Medicine and Animal Production, University of Naples “Federico II”, 80137 Naples, Italy; chiara.caterino@unina.it (C.C.); giovanni.dellavalle@unina.it (G.D.V.); stefano.cavalli@unina.it (S.C.); jacopo.guccione@unina.it (J.G.); francesco.lamagna@unina.it (F.L.); fatone@unina.it (G.F.)

**Keywords:** PRP, cattle, platelet-rich plasma, regenerative medicine, growth factors, mastitis, uterine dysfunctions, ovarian dysfunctions

## Abstract

**Simple Summary:**

Platelet concentrates have become increasingly popular in both animal and human medicine. These concentrates mainly consist of higher-than-normal levels of platelets and therefore contain a high concentration of growth factors which are stored within platelet α-granules. One specific type of platelet concentrate, known as Platelet-Rich Plasma (PRP), is notable for its low-density fibrin. Recent research has focused on how PCs can benefit cattle, and PRP has shown promise in treating various reproductive issues, assisting in the production of bovine embryos, and addressing problems like sole ulcers and udder diseases. In this report, we aim to review the existing literature on the use of PRP in bovine veterinary practice. We followed strict guidelines to identify relevant studies. After searching extensively through databases, we found a total of 1196 papers, but only six met our criteria for inclusion. Out of these six papers, two discussed the use of PRP for mastitis, two for uterine dysfunction, and two for ovarian dysfunction in cattle. PRP emerged as a cost-effective and readily available treatment option, showing positive results in these cases. However, due to the diversity of conditions and definitions involved, more research is needed to fully understand its potential in clinical practice for cattle.

**Abstract:**

Platelet concentrates (PCs) have become widely used in veterinary and human medicine. The PCs consist mainly of supraphysiological concentrations of platelets and, therefore, growth factors (GFs) which are stored within platelet α-granules. Among PCs, Platelet-Rich Plasma (PRP) is characterised by low-density fibrin. Research on the effect of PCs in cattle has surged in recent years; in particular, evidence has shown the positive use of PRP for treating reproductive problems, in vitro production of bovine embryos, sole ulcers and udder diseases. The aim of this report is to critically review, in accordance with the PRISMA guidelines, the available literature reporting clinical application in the bovine practice of PRP. Three bibliographic databases PubMed, Web of Science and Scopus were used for a broad search of “platelet concentrates” OR “PRP” OR “platelet-rich plasma” OR “PRF” OR “platelet-rich fibrin” AND “cows” OR “cattle”. From 1196 papers, only six met the inclusion criteria. Two papers described the use of PRP in mastitis, two papers in uterine dysfunction and two papers in ovarian dysfunction. PRP offered a low-cost, easily obtained therapeutic option and showed positive results for these patients. However, given the different pathologies and definitions involved, further studies are necessary to assess its full clinical potential.

## 1. Introduction

Platelet concentrates (PCs) are biological autologous blood products that have been used since their introduction in dental and oral surgery [1], sports medicine and orthopaedics [2,3], reproductive affection [4,5,6] mastitis [7,8]. The PCs consist mainly of supraphysiological concentrations of platelets and, therefore, growth factors (GFs) stored within platelet α-granules [9]. Based on a classification commonly used in human medicine [10], PCs are divided into two main categories based on fibrin architecture: platelet-rich plasma (PRP) with a low-density fibrin network; and platelet-rich fibrin (PRF) with a high-density fibrin network. These PCs can be further classified according to the presence of leukocytes into Pure-PRP (P-PRP) and Leukocyte-PRP (L-PRP) without and with leukocytes, respectively (Figure 1). In the same way, Pure-PRF (P-PRF) and Leukocyte-PRF (L-PRF) are characterized by the absence, in the former, and the presence, in the latter, of leukocytes [9]. Another important difference between PRP and PRF families is the use of anticoagulated blood for PRP preparation, whereas PRF is prepared from non-anti-coagulated blood. PCs are prepared using gravitational centrifugation techniques, standard cell separators, or autologous selective filtration techniques [11]. Research into the effects of PCs in cattle has increased considerably in recent years; in particular, evidence has shown the positive use of PRP for treating reproductive problems and in vitro production of bovine embryos [4,5,12,13,14]. A recent study has shown the efficacy of PRP in reducing the use of antibiotics in subclinical mastitis [8]. Tsuzuki et al. [15] described the positive regenerative effects of the use of gelatine microspheres incorporated with PRP and alginate in sole ulcers and has a beneficial effect on articular cartilage and ligament healing [16,17,18]). Moreover, the effectiveness of PRP has also been evaluated in in vitro studies regarding the chondroprotective effects of autologous PRP combined with antibiotics in septic arthritis caused by Staphylococcus aureus [19]. Regarding GFs, the amount released by PRP and the comparison of the concentrations of platelets and white blood cells in PRP’s layers and supernatant have been described [20], but the comparison of GFs release over time has not been investigated yet, as well as their exact antimicrobial role [19].

While numerous studies have investigated the in vitro effects of PRP [13,14,19,21], translating these findings into effective in vivo applications remains a subject of ongoing research and debate. The present study seeks to address this gap in the literature by systematically reviewing the existing body of knowledge regarding the in vivo utilisation of bovine PRP (PRP) within the same specimens.

## 2. Materials and Methods

### 2.1. Search Strategy

The systematic review followed the PRISMA (Preferred Reporting Items for Systematic Reviews and Meta-Analyses) guidelines [22].

The time period in which the database was searched for relevant publications was not limited to the period from May 2023 onwards. A thorough literature search focused on platelet concentrates application in in bovine practice, encompassing studies published in English, Spanish, and Italian languages. Four authors (CC, GDV, FA, and SC) were responsible for the research, and any disagreements were resolved through discussion or, if necessary, by a third author (GF).

### 2.2. Inclusion Criteria

Inclusion criteria for the search involved the presence of relevant terms such as “platelet concentrates” “PRP”, “platelet-rich plasma”, “cows”, “cattle”, “PRF”, and “platelet-rich fibrin” in the full manuscript, abstract, title, and keywords of publications indexed by web search engines that provide access to full-text articles (e.g., Pubmed, Web of Science, and Scopus). In addition, the bibliography of the selected papers was critically reviewed in order to improve the sources. Case reports, retrospective or prospective case series including a sound description of centrifugation protocol, clinical application and outcomes were also considered as inclusion criteria.

### 2.3. Exclusion Criteria

Studies were excluded from the comprehensive literature review if they centred on species other than cattle, utilized alternative biomaterials or substances in combination with platelet concentrates, or pertained solely to in vitro research.

Due to their limited availability, clinical trials were incorporated irrespective of their level of evidence or design, without discrimination between studies featuring a control group and those lacking one.

### 2.4. Study Selection

The initial review involved examining the titles and abstracts of the first 100 entries and any discrepancies were resolved through discussion until a consensus was reached. Subsequently, two researchers (CC and FA) independently assessed the titles and abstracts of all articles found. In case of disagreement, a consensus was reached through discussion on which articles should be selected for full-text review. If necessary, a third researcher (GF) was consulted for the final decision. Two researchers (GDV and SC) then independently reviewed the full texts to determine the suitability of the articles. Again, any disagreements between the authors were resolved, with the option of involving a third reviewer (JC).

### 2.5. Data Extraction

Two authors (CC and GDV) conducted data extraction independently. In case of disagreements, they were resolved through discussion or consulting with another author (GF). Data extracted included: year of publication, authors, journal of publication, type of platelet concentrates used (PRP, P-PRP etc.), centrifugation protocol used, autologous or allogenic platelet concentrates used, site of blood collection, breeds, studies classification such as in vivo or in vitro studies, randomized controlled trials (RCTs) or not (No-RCTs), sample size, outcome measurements (which were categorised as positive, negative or neutral) and adverse events.

Synthesis Without Meta-analysis (SwiM) checklist [23] was applied, because of the inclusion of different study designs and applications. Meta-analyses could not be performed due to the heterogeneity of the study designs. 

FA and FL independently applied the Joanna Briggs Institute (JBI) Critical Appraisal Checklists to evaluate the quality of papers enrolled. Every study design was combined with the corresponding JBI checklist. The checklist for qualitative research comprised a total of 10 sub-items. In this systematic review, we reviewed the qualitative research checklists to increase the accuracy of the assessment. If it was clearly described, it was rated as “Yes”, “No” if it was not presented, “Unclear”, if it was not clear, and “Not applicable” if it could not be applied. All discrepancies were resolved through discussion.

## 3. Results

### 3.1. Database Review

A total of 1196 papers were found in the databases mentioned above. A flow chart of the literature search and study selection is shown in Figure 2.

After the removal of duplicates, we reviewed 1065 manuscripts. Only 6 articles met the inclusion criteria; selected studies are reported and numbered in Table 1.

Four studies of the six included [4,5,6,12] described in vivo application of bovine PRP in reproduction problems and the remaining two the application of the platelet concentrate for the treatment of mastitis [7,8] 

Four studies used Holstein-Friesian cows [4,5,6,12]; while, one study [7] did not specify the breed of the animals. Finally, only a study used creole Colombian bovine breed (Blanco Orejinegro) [8].

Regarding the side of blood collection, five out of seven studies [4,5,6,7,12] indicated the subcutaneous mammary vein; on the contrary, jugular vein collection was selected by one study (8).

Most of the trials studied the effects of PRP obtained from donor cows [4,7,8,12] two studies analysed the effect of homologous PRP [5,6].

No side effects have ever been reported following PRP administration.

### 3.2. Centrifugation Protocols

Five of the six studies included followed the double-centrifugation method [4,5,6,7,12] which was first described by Lange-Consiglio [7]. The method consisted of whole blood sampling and storing into blood collection bags containing citrate-phosphate dextrose-adenine (CPDA-1). The blood was then transported to the laboratory at +4 °C within 2 h of collection and processed immediately. Next, whole blood was centrifuged at 100× *g* for 30 min to allow the separation of the three components of the blood: red blood cells at the lowest level, “buffy coat” comprised in the middle, and PRP in the upper layer. The PRP is then further centrifuged at 1500× *g* for 10 min to obtain platelet pellet and poor platelet plasma (PPP) in the upper layer. Subsequently, two-thirds of the volume of PPP was aspirated for later use and the pellet was mixed into the remaining PPP volume to allow platelet counting prior to final dilution with PPP to obtain PC at a standard concentration of 1 × 109 platelets/mL.

Only one of 6 studies [8] performed a different protocol aimed to obtain P-PRP for intramammary use. Briefly, the blood bag was centrifuged at 1600× *g* for 8 min. The plasma fraction (P-PRP), including the buffy coat of each blood bag, was then transferred to a separate plasma bag. Before the intramammary infusion, calcium gluconate was added to the P-PRP in order to induce platelet activation and GFs release.

### 3.3. Mastitis

In 2014, Lange-Consiglio and colleagues [7] considered cows as affected by chronic mastitis when the somatic cells count (SCC) measured >500,000 cells/mL for three consecutive months as previously described [24]; while cows were affected by acute clinical mastitis when there were visible signs such as abnormal milk, hot, swollen and painful quarter [25,26]. For the treatment of mastitis, in order to assess the effect of PRP in the treatment of both forms of mastitis, included animals were randomly allocated in three groups. The first group received 5 mL of PRP alone, the second group received 5 mL of PRP added to an antibiotic chosen based on an antibiogram test and, the third group received and the antibiotic alone. Regardless of the group, treatment was administered for three consecutive days. The number of SCC was then assessed before starting the treatment and reevaluated at 7, 14 and 30 days after it. In addition, clinical and laboratory evaluations such as decreasing by at least 3 linear score (LS) points of SCC compared with the starting value and macroscopic aspects of milk and udders were considered in order to assess the efficacy trends. The rate of relapse rate, namely a clinical mastitis event that occurred in the analysed quarter between the last time of investigation (30 days after treatment) and the end of the same lactation, was further evaluated. The results showed that PRP and antibiotic had significantly superior performance since the mean value of LS decreased more significantly than the other treatments, but PRP alone gave better results than antibiotic alone in chronic mastitis. According to the authors, these results could be a direct consequence of the release of growth factors from platelets derived by PRP which could protect and enhance the healing process of the damaged tissue. Moreover, PRP produced better results regarding relapse.

On the other hand, Duque-Madrid and colleagues [8] considered cows affected by subclinical mastitis (SCM) (27). According to the definition used by the authors [27], subclinical mastitis is characterized by the absence of clinical signs of mammary gland inflammation, and is generally associated with high SCC number. In addition, the inclusion criteria included the absence of blind mammary quarters, no history of chronic mastitis in the current lactation and neither administration of any antibiotics for any other disease in the 30 days before enrolment. The set cut-off value for SCC was ≥100,000 cells/mL in primiparous cows and ≥200,000 cells/mL in multiparous cows; while intramammary infection was diagnosed when a composite milk sample showed a higher SCC than the cut points and the microbiological culture was positive for any major Gram-positive mammary pathogen. The authors used 10 mL of activated P-PRP for one group at 12 h intervals and, a conventional broad-spectrum fourth-generation cephalosporin (75 mg) administered during three consecutive milking in the control group. The SCC number was assessed before starting the treatment and at 21, 22, 47 and 48 days after it. In addition, the changes in cytokines in the milk (day 0, 21 and 47) were used as criteria for assessing the inflammatory status of the mammary gland. The authors concluded that conventional broad-spectrum antibiotic showed better antimicrobial response compared to P-PRP. Moreover, the study’s results showed a significantly increased value of mean the LnSCC only in the P-PRP group but, an increase in the concentration of milk cytokines in both groups.

### 3.4. Uterine Dysfunctions

In 2015 Lange-Consiglio and colleagues [4], in a 3-step study (in vitro, immunohistochemically and in vitro), aimed to evaluate the effect of PRP in repeat breeder cow (RBC).

According to the authors uterine application of PRP may be useful in peri-implantation or in the healing process of clinically silent endometrial lesions, as cytokines act as links in implantation, placentation and foetal growth. To assess these points, authors evaluated different concentrations of PRP on embryo development and the immunohistochemical analysis of the endometrium after the administration in vivo. Moreover, the rate of embryo implantation in treated cows was evaluated 48 h after the artificial insemination (AI). The treatment consisted of 10 mL of PRP on the ninth day of the dioestrus stage: in two cows it was injected in the uterine horn ipsilateral to the corpus luteum (maintaining the contralateral horn as control), and in the other two cows PPR was administered in the horn contralateral to the corpus luteum.

In the in vivo step of the study, two groups each of 30 cows were randomly selected to receive AI and PRP or AI alone using the semen from the same bull. Ultrasonographic evaluation at day 32 and day 60 was performed to assess the pregnancy status. The study’s results showed an increased number of proliferating nuclei in the treated horn, no macroscopical alteration of the uterine epithelium between the groups, and a pregnancy status rate of 70% in the AI+PRP group and 30% in AI group. The authors assumed that the platelet growth factors’ release, and in particular Epidermal Growth Factor (EGF), increased significantly the pregnancy status rate.

In 2016, Marini and colleagues [12] investigated the potential use of PRP for the treatment of endometritis in cows with an in vivo and two in vitro experiments. The risk of uterine contamination in the first two weeks after parturition reaches 80–100% [12] but only in 10–20% of the cows the infection is uncontrolled and leads to chronic uterine inflammations [28,29,30,31,32]. In particular, E.coli which produces an endotoxin lipopolysaccharide (LPS) is one of the main pathogenic bacteria responsible for this infection [33]. The uterine infection leads to a decreased pregnancy status rate. Therefore, since platelets included in PRP are able to release growth factors and anti-inflammatory cytokines [34], the authors evaluated the in vivo effect of PRP through the histological and immunohistochemical examinations of endometrial biopsies obtained just before treatment (day 4 post-estrus-T0) and 7-day post-treatment (11 days post-estrus-T1). For this purpose, 14 cows were randomly enrolled and divided in two groups, one receiving 10 mL PRP four days after oestrus, and the other stated as the control group received 10 mL of 0.9%NaCl. In addition, the immunohistochemically detected expression of the progesterone receptor (PR) was used as a marker for the susceptibility of the cells [35]. No pathological changes were detected in any of the biopsies examined and PR expression showed a similar average value in the epithelial cells of all animals at T0 and T1. Nevertheless, PR expression at T1 was increased in glandular epithelial and smooth muscle cells in the PRP animals, suggesting a possible role in maintaining and/or increasing PR numbers in uterine tissue.

### 3.5. Ovarian Dysfunctions

Both Cremonesi’s studies focused on the intraovarian administration of PRP, the former in order to treat ovarian hypofunction [5], and the latter to improve embryo recovery [6]. According to the authors, ovarian hypofunction was defined by the presence of follicles with a diameter of at least 8–15 mm visible in two consecutive examinations, by the absence of a corpus luteum or cysts without signs of oestrus in the 7-day period between the examinations [36].

In the first study, 12 cows with a diagnosis of ovarian hypofunction and in anoestrus were enrolled and randomly allocated in two group: one group received 5 mL of PRP, while the other group stated as control group received 5 mL of placebo (0.9% NaCl). The injection was performed under ultrasound guidance. Blood samples were taken in both groups before intraovarian injection (T0) and after 2 and 4 weeks to assess the progesterone serum concentration. The post-treatment was assessed by ultrasonography to detect follicles and corpus luteum. The study’s results showed the presence of follicles and corpus luteum only in the PRP group. When the oestrus status was detected, the AI with frozen semen was performed. According to the authors’ results, the pregnancy status rate in the treatment group was 100%. In four out of five of these animals, a mild to strong progesterone increase was detected while no variation in progesterone was detected in the control group.

The second study [6] evaluated the intraovarian effect of autologous PRP before superovulation in order to increase the number of follicles responsive to gonadotropin treatment and, as a consequence, improve embryo recovery in donor cows. Eight cows which calved normally between 80 and 90 before the enrolment, had the synchronisation of the oestrus cycle. Daily ultrasonographic monitoring for the consecutive cycles was performed. All cows underwent intra-ovarian PRP injection. The right ovary of each cow was left as a control, while the left ovary was injected with 5 mL of PRP (at a concentration of 1 × 109 platelets/mL). After 9 days, the corpus luteum and ovarian follicular status were detected and superovulation was induced with 1000 international units (IU) of a commercial gonadotropin preparation (Gn) (500 IU FSH + 500 IU LH) twice daily for 5 days at 12-h intervals. Seven days after AI, the uterine horns were flushed, and the embryos were collected. The average count of follicles, as determined by transrectal ultrasound examination, did not display a statistically significant variance between the treated and control ovaries (*p* = 0.28). Likewise, there was no notable difference observed 48 h after the injection of Platelet-rich Plasma (PRP) (7.67 ± 2.52 and 8.00 ± 2.00, *p* = 0.73, respectively). However, a statistically significant distinction (*p* = 0.023) emerged in the mean follicle count at the final gonadotropin injection between the control and treated ovaries. As per the study findings, the authors propose that intra-ovarian administration of PRP before superovulation might yield beneficial effects on both the development of dormant follicles and the production of embryos in vivo.

## 4. Discussion

This paper reviews and summarises the current evidence of the clinical application of PRP in bovine practice through a systematic review of available literature. Over the past two decades, there has been a growing trend in the utilisation of blood derivatives in several medical scopes [1,2,3]. The rationale for employing platelet concentrates lies in their capacity to release growth factors produced and secreted by the enclosed platelets [3,34,37]. Similar to human medicine, PRP stands out as the most widely employed blood product in veterinary medicine. Pure Platelet-Rich Plasma and Leukocyte- and Platelet-Rich Plasma are liquid platelet suspensions, respectively, without and with leukocytes [38]. They can be used as injectable suspension leading to wide application in medical fields. Several PRP production protocols have been described, varying based on separation procedures, the use of anticoagulants, or activators. The centrifugation rate has been demonstrated to be a crucial factor in determining the ideal platelet yield [39]. In the majority of PRP production processes, two centrifugation steps are employed, one for separation and the other for concentration. Most of the selected studies (5 out of 6) [4,5,6,7,12] employ the same centrifugation protocol; therefore, it is hard to compare quantitative variations in the obtained concentrates critically. Duque-Madrid [8] activates the PRP with calcium gluconate before clinical use after obtaining it. Usually, when introducing PRP into connective tissues, it encounters collagen and tissue thromboplastin (tissue factor), which can trigger platelet activation [40]. Consequently, the necessity for platelet activation using exogenous thrombin before injection remains uncertain. Furthermore, Han and colleagues indicated that PRP’s activation with exogenous thrombin might, in reality, have a detrimental impact on the biological activity of the growth factors [41]. Moreover, other authors [42] have reported that thrombin activation allows for an immediate release of growth factors. However, the inactivated PRP probably also releases growth factors, but much more slowly, so that the cells receive a lower sustained dose overall [41].

Regarding the clinical evidence of PRP in bovine practice, no data were found about soft tissue trauma, teat trauma, and ocular lesions. A few of the reported studies focus on mastitis treatment, and reproductive disorders, with specific attention given to hypofertility associated with ovarian and uterine issues. There are several risk factors that are known to be associated with the occurrence of bovine mastitis and play an important role, including pathogen, host and environmental factors [43]. As a definition, bovine mastitis is an inflammatory reaction of the udder tissue in the mammary gland resulting from physical trauma or infection with microorganisms. It is the most common disease of dairy cows and causes economic losses due to reduced milk yield and poor milk quality [44]. he aetiological agents include a variety of Gram-positive and Gram-negative bacteria and can be infectious (e.g., *Staphylococcus aureus*, *Streptococcus agalactiae*), environmental (e.g., *Escherichia coli*, *Enterococcus* spp.) or opportunists (e.g., Coagulase-negative Staphylococci) [45] Aside from reduced production, economic losses related to clinical mastitis encompass the expenses for treatment, additional labour, and an elevated cow replacement rate [46]. It is important to recognize that dairy cows produce milk for human consumption, making mastitis a food safety concern as well [47]. A proficient and productive mastitis control program necessitates several key components. These include early infection detection through an understanding of pathogenesis, the development of sensitive diagnostic tests for early screening, the implementation of sound management practices to minimize transmission risks, and the protection of uninfected individuals. Additionally, the control program must encompass a strategic approach to antimicrobial usage to mitigate concerns related to antibiotic residues in milk and the development of antimicrobial resistance [48]. Consequently, the dairy industry has undertaken measures to curtail antibiotic usage. As a result, there is a growing interest in exploring alternative approaches for the prevention and treatment of bovine mastitis, with a specific focus on natural products derived from botanical and animal sources [49]. Lange-Consiglio [7] and Duque-Madrid [8] both studied the application of PRP in this field. The former author considered cows affected by acute and chronic mastitis, while the latter cows with a diagnosis of subclinical mastitis and no history of chronic mastitis. In order to be able to compare the results of the two studies, despite considering pathologies of a different degree, it is important to consider that, contagious agents are predominantly associated with subclinical infections [43,50], and therefore dry cow therapy (DCT) and long persisting antibiotics, as they can deliver better cure rates [51,52]. While, in certain extreme acute mastitis, when inflammation is particularly severe and complete milk extraction becomes challenging, milk ducts may become obstructed by milk residue. This obstruction can impede the distribution of antibiotics and other medication (presumably even PRP) throughout the udder [53]. This concept, associated with the possible role of growth factors in tissue protection or repair processes, or both [54] could explain why PRP appeared to be more effective in chronic mastitis rather than in acute and subclinical ones.

Endometrial receptivity plays a pivotal role in the successful implantation of embryos. The economic ramifications of this disease are significant, encompassing a substantial impact on the economy through reduced milk yield, calf production, and treatment expenses [55]. According to the latest definition, metritis is a disease characterised by an enlarged uterus and a watery, reddish-brown to viscous, whitish-purulent discharge from the uterus, which often has a foetid odour. Metritis is most common within 10 days of parturition [28]. It can be further classified in three grades based on the presence of systemic signs of illness and toxaemia [28]. Finally, endometritis can have clinical and subclinical appearance: the former is characterized by the presence of purulent (>50% pus) or mucopurulent (approximately 50% pus, 50% mucus) uterine exudate in the vagina, 21 days or more postpartum, and is not accompanied by systemic signs; while the latter it is defined by >18% neutrophils in uterine cytology samples collected 21–33 days postpartum, or >10% neutrophils at 34–47 days in the absence of clinical endometritis [28]. The reported treatment consists mainly of intrauterine antimicrobials [56,57,58] and active prevention Unfortunately, treatment efficiency has not been completely proved for endometritis and nowadays is mandatory to reduce the use of antimicrobial products particularly in food-producing animals and to look for alternative treatments. Lange-Consiglio et al. [4] and Marini et al. [12] described the use of PRP for clinically silent endometrial injuries in repeat breeder cows and uterine infections, respectively. Both studies achieved good results obtaining a higher pregnancy rate in the treated group [4] and higher PR expression [12], but there was no consensus regarding the definition of endometritis in the considered study. In particular Marini and colleagues [12] considered sound cows, therefore it could be challenging to evaluate the clinical impact on animals. Studies in humans [59,60,61] have shown that cytokines and growth factors could improve embryo implantation. The same growth factors, such as Fibroblast Growth Factor, Transforming Growth Factor-6, Platelet-Derived Growth Factor (PDGF), and Epidermal Growth Factor (EGF) [62], which have the potential to stimulate bovine embryo development [63,64]. Furthermore, endometrial cells naturally produce EGF in vivo, and its receptors have been identified in the embryo [65]. As pregnancy progresses, the increased production of EGF plays a role in enhancing trophoblast differentiation, which is critical for cell attachment and embryo development [65]. It was observed that the treated group had a higher number of pregnant cows compared to the control group. This observation would imply that the intrauterine administration of platelet concentrates 48 h after insemination, a timing chosen to minimize disruption to spermatozoa progression and before the embryo reaches the uterus, might create a conducive uterine environment for embryo implantation. This favourable effect could be attributed to the influence of platelet growth factors on the regeneration and potential healing of clinically silent endometrial irregularities [66]. Marini also concurred with Lange Consiglio and demonstrated the anti-inflammatory and stimulatory effects of PRP on endometrial cells. Additionally, the introduction of PRP is expected to enhance the uterine environment by providing the essential factors required for embryo development. Uterine glands produce a secretion rich in glycoproteins, which is believed to play a supportive role for the embryo during the pre-implantation period.

The rationale behind the use of PRP in ovaries dysfunctions, lies of the release of growth factors which has been described as an effective treatment in humans poor ovaria reserve [67,68]. Therefore, Cremonesi et al. in both his studies [5,6] aimed to demonstrate the feasibility of this procedure even in bovine species. The effect of PRP on ovarian dysfunction is presumably due to the secretion of nutrients that have a positive effect on the follicles. In addition, as shown in human medicine PRP would increase the dimensions of the follicles [69,70] due to the release of Platelet-Derived Growth Factor, the Transforming Growth factor-β (TGF-β), and the Hepatocyte Growth Factor which have mitogenic or trophic effects and, specifically, the TGF-β that can stimulate cell proliferation and differentiation. While the increase of progesterone was a marker of restored ovarian cycle. In Cremonesi et al. [6], even if it was carried out on healthy cows, PRP could distribute mediators inside the ovary and regulate angiogenesis, ovarian blood flow and oocyte competence [68]. Another hypothesis is that growth factors can communicate with ovarian stem cells and induce their differentiation in ex novo oocytes.

## 5. Conclusions

In conclusion, scarce data are available regarding the evidence of clinical use of PRP in bovine practice. Nevertheless, it can be considered as an alternative therapeutical approach to treating and managing specific pathologies, especially in animals’ refractory to standard therapy. It offers a low-cost, easily obtained and reliable therapeutic option to these patients. Blood sampling is a standard procedure and the autologous features of the treatment eliminate the risk of viral or bacterial transmission between animals. However, the main targets in bovine practice reported are the udder and the reproductive system. Consequently, the lack of scientific data based on strong evidence about its use in other clinical scenarios makes high-quality research to standardize case definition, and possible applications crucial to expand knowledge.

## Figures and Tables

**Figure 1 vetsci-10-00686-f001:**
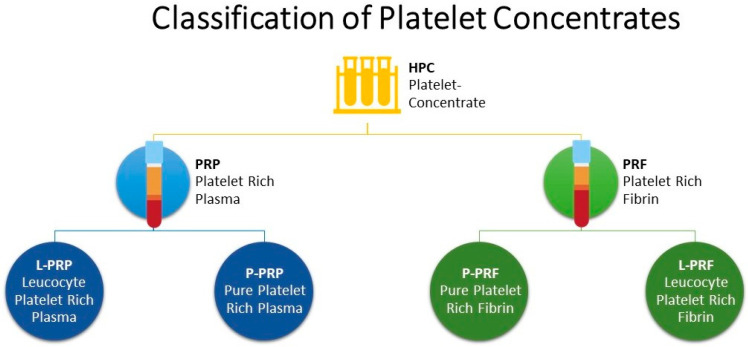
Classification of Platelet Concentrates.

**Figure 2 vetsci-10-00686-f002:**
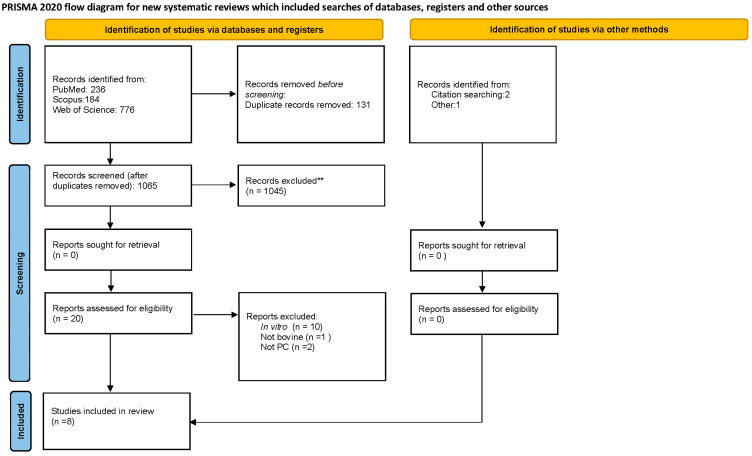
PRISMA 2020 flow diagram for new systematic reviews included searches of databases, registers, and other sources.

**Table 1 vetsci-10-00686-t001:** Study, sample constitution, interventions applied, and outcomes considered in the studies involved in the systematic review.

Authors	Year	N. Cases	Breed	Site of Blood Collection	Centrifugation Protocol	Aim	Outcome	JBI
Lange-Consiglio	2014	229	Not specified	Subcutaneous mammary vein	Double centrifugation method	Evaluate the effect of PRP, alone or associated with antibiotic in the control of clinical acute and chronic mastitis	PPR and antibiotic showed significantly superior performance but PRP alone showed better results in chronic mastitis	Yes, include
Lange-Consiglio	2015	65	Holstein-Friesian	Subcutaneous mammary vein	Double centrifugation method	The in vivo experiment evaluate embryo implantation and development in repeat breeder cows after intrauterine administration of PRP at 48 h following artificial insemination	administration of PPR 48 h after AI, it is supposed to be the ideal time in order to not disturb spermatozoa progression and before the embryo reaches the uterus, making the uterus more favourable to embryo implantation	Yes, include
Marini	2016	14	Holstein-Friesian	Subcutaneous mammary vein	Double centrifugation method	Evaluate the effect of PRP in vivo and in vitro, on a model of healthy (in vivo) or endotoxin lipopolysaccharide stressed bovine endometrial cells	PRP might be helpful in maintaining and/or increasing the number of progesterone receptors	Yes, include
Cremonesi	2020	8	Holstein-Friesian	Mammary vein	Double centrifugation method	Evaluate if administration of autologous PRP inside the bovine ovary before gonadotropin treatment could increase the number of follicles responsive to superovulation in order to improve embryo recovery from eight donor cows	Results show that after the superovulation protocol, a statistically significant difference (*p* ≤ 0.05) was detected between PRP treated and control ovaries	Yes, include
Cremonesi	2020	12	Holstein-Friesian	Subcutaneous mammary vein	Double centrifugation method	Evaluate the effect of PRP in restoring ovarian function in cows with ovarian failure through the measurement of progesterone levels as a marker for restored ovarian cycling and the monitoring of oestrus to perform AI	after intraovarian PRP administration (week 0), the level of PRG increased between 3- and 11-fold during the subsequent 4 weeks	Yes, include
Dunque-Madrid	2021	103	Creole Colombian bovine	Jugular vein	1600 g × 8 min	Compare the cure risk of intramammary treatment of pure platelet rich plasma (P-PRP) or cefquinome sulphate (CS) in cows with subclinical mastitis caused by Gram-positive bacteria, evaluated via somatic cell count and the microbiological analysis of milk; to compare the inflammatory/anti-inflammatory response of mammary gland to both treatments through the analyses of interleukins (IL), interferon gamma (IFN-γ), and tumour necrosis factor alpha (TNF-α) in milk	The group of SCM cows treated with PRP presented a lower rate of bacteriologic cure when compared to SCM animals treated with CS	Yes, include

## Data Availability

Not applicable.

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
