# Peer review of "Clinical Application of Platelet Concentrates in Bovine Practice: A Systematic Review"

_vetsci, 2023, doi:10.3390/vetsci10120686_

Round 1

Reviewer 1 Report

Comments and Suggestions for Authors

The work follows a logical approach and effectively links data, bibliography and results. The PRISMA flowchart ensures documentation of all stages of the review process.
  This work is interesting because, given numerous publications, it demonstrates the importance of scientific data analysis for the safe use of health protocols.

Author Response

Thank you very much for taking the time to review this manuscript.

Reviewer 2 Report

Comments and Suggestions for Authors

Author Response

Thank you very much for taking the time to review this manuscript.

Clarity and scope: Thank you for pointing this out. We have added some clinical implications of this research in the discussion.

Methodology and Technical Details: We agree with this comment and you can see the lines 85-100 for the inclusion and exclusion criteria.

Results and Discussion: we agree and have revised the text accordingly to emphasize this point.

Relevance and Clinical Applicability: Thank you for your comment. We have highlighted this point in the Discussion and Conclusions.

Reviewer 3 Report

Comments and Suggestions for Authors

Caterino et al.'s work is a review that describes the state of knowledge of the application of platelet concentrates in the bovine clinic.

The work describes the results of six works that meet the inclusion criteria.

The works describing the use of platelet concentrates in the bovine clinic are limited, and the therapeutic efficacy of these therapies has yet to be demonstrated.

The work is well written. The English language is clear and understandable; there are only minor typos or spelling errors.

The content of the review gives a sound idea of the application possibilities of these therapies.

However, some doubts arise on the comparative description of the works dealing with mastitis treatment, i.e., the works from  Lange-Consiglio (7) and Duque-Madrid (8). The two works use substantially different PRP preparations in terms of platelet content. Much higher in (7), somewhat limited in (8): in (8), the platelet content is similar to that of the starting blood. No platelet concentration occurs, indeed. I believe these aspects should be highlighted to the reader when dealing with platelet concentrates review. Furthermore, the paragraph relating to the "Discussion" does not indicate that the clinical outcome reported in (8) is substantially inferior to antibiotic treatment. The data is reported in paragraph 3.3, but it should be better highlighted and discussed in the final discussion.

Finally, nowhere does the work indicate numerical data relating to the definition of platelet concentrate. In paragraph 1.Introduction, the various classification parameters proposed for platelet concentrates are mentioned, but no reference is made to the meaning of "supraphysiological concentration of platelet". The literature extensively discusses how PRP can be classified based on platelet content, which should be introduced in the manuscript.

Comments on the Quality of English Language

English language is correct. Only minor typing errors are present.

Author Response

As corresponding, the authors (AU) thank the Reviewers for the important and valuable comments that helped to improve the quality of the manuscript. The AU have carefully considered every reviewers’ recommendation and the manuscript has been revised according to their suggestions.

Your Sincerely

Dr. Federica Aragosa

Caterino et al.'s work is a review that describes the state of knowledge of the application of platelet concentrates in the bovine clinic.

The work describes the results of six works that meet the inclusion criteria.

The works describing the use of platelet concentrates in the bovine clinic are limited, and the therapeutic efficacy of these therapies has yet to be demonstrated.

The work is well written. The English language is clear and understandable; there are only minor typos or spelling errors.

The content of the review gives a sound idea of the application possibilities of these therapies.

However, some doubts arise on the comparative description of the works dealing with mastitis treatment, i.e., the works from  Lange-Consiglio (7) and Duque-Madrid (8). The two works use substantially different PRP preparations in terms of platelet content. Much higher in (7), somewhat limited in (8): in (8), the platelet content is similar to that of the starting blood. No platelet concentration occurs, indeed. I believe these aspects should be highlighted to the reader when dealing with platelet concentrates review. Furthermore, the paragraph relating to the "Discussion" does not indicate that the clinical outcome reported in (8) is substantially inferior to antibiotic treatment. The data is reported in paragraph 3.3, but it should be better highlighted and discussed in the final discussion.

Finally, nowhere does the work indicate numerical data relating to the definition of platelet concentrate. In paragraph 1.Introduction, the various classification parameters proposed for platelet concentrates are mentioned, but no reference is made to the meaning of "supraphysiological concentration of platelet". The literature extensively discusses how PRP can be classified based on platelet content, which should be introduced in the manuscript.

AU: Okay, the manuscript has been accordingly modified.

Reviewer 4 Report

Comments and Suggestions for Authors

The authors present a systematic review regarding the use of platelet concentrates in cattle.

The manuscript offers a useful guide in the topic and can be published after some minor changes, as outlined below.

-For only 6 references within the remit of the study topic, the manuscript is rather long (16 pages) and should be significantly reduced in length. The excessive length really shadows the heart of the manuscript, which is the review of these 6 references. I believe that a manuscript 10-page long will do the work just as well and will tire the readers with excessive and verbose paragraphs.

-The authors should add ‘Bos taurus’ in the search string to improve the results of the search. Also, the search can be update until say 20 November 2023, in order to have a complete and fully update view of the relevant literature.

Author Response

As corresponding, the authors (AU) thank the Reviewers for the important and valuable comments that helped to improve the quality of the manuscript. The AU have carefully considered every reviewers’ recommendation and the manuscript has been revised according to their suggestions.

Your Sincerely
Dr. Federica Aragosa

Comments to the Author

Reviewer #4

The authors present a systematic review regarding the use of platelet concentrates in cattle.
The manuscript offers a useful guide in the topic and can be published after some minor changes, as outlined below.

-For only 6 references within the remit of the study topic, the manuscript is rather long (16 pages) and should be significantly reduced in length. The excessive length really shadows the heart of the manuscript, which is the review of these 6 references. I believe that a manuscript 10-page long will do the work just as well and will tire the readers with excessive and verbose paragraphs.

-The authors should add ‘Bos taurus’ in the search string to improve the results of the search. Also, the search can be update until say 20 November 2023, in order to have a complete and fully update view of the relevant literature.

AU: Thank you for your comments. We have discussed the length of the manuscript with the editor; unfortunately, the tables take up multiple pages. The final editing will address to reduce the length of the manuscript.
